# Convolutional neural network (CNN) based month's name recognition in Gurumukhi script for Punjab state of India

Tajinder Pal Singh[1], Sheifali Gupta[1], Jamil Hussain[2], Sapna Juneja[3], Meenu Garg[1], Deepali Gupta[1] and Majed Alsafyani[4]

[1] Chitkara University Institute of Engineering and Technology, Chitkara University, Rajpura, Punjab, India
[2] Department of Artificial Intelligence Data Science, Sejong University, Sejong, Republic of South Korea
[3] Department of CSE(AI), KIET Group of Institutions, Ghaziabad, India
[4] Department of Computer Science, College of Computers and Information Technology, Taif University, Taif, Saudi Arabia



## ABSTRACT

In the context of natural language recognition, the development of an automated system for data analysis and interpretation has tremendous demand. Developing such kinds of systems for a country like India, however, has been found to be a laborious task when compared to other countries due to India's use of multiple scripts and languages. Gurumukhi is the regional language of Punjab, and the development of an automated text recognition system in the Gurumukhi language is found to be very critical because of the character's intricate structure. Influenced by this problem, the present work has been conducted to design an error-free classification model of Gurumukhi text. A convolutional neural network (CNN) based classification model was designed to perform holistic word recognition of Gurumukhi months. For this work, a dataset of 24,000 word images of 24 different Gurumukhi months has also been prepared from 500 distinct writers. The highest accuracy obtained using the proposed model on validation data is 99.7%.

## INTRODUCTION

In the context of natural language processing, research on handwritten text recognition in regional languages is on trend. Text recognition helps to analyze or recognize the data in records (*Garg, Jindal & Singh, 2019*). Error-free text recognition can aid in developing an automated text reading and analysis system for various applications such as postal automation, automated bank checks, and ID proofs, reading systems, *etc*. Developing such kinds of systems for a country like India, however, has been found to be a laborious task when compared to other countries due to India's use of multiple scripts and languages. Gurumukhi is a complex script due to the structure of its characters, and the same has been considered for recognition in the present work as the Punjab government has designated it as the official language. Previously, many models have been proposed in Gurumukhi text for character recognition and word recognition. All such models relayed on types of methods used for feature extraction on text images for their performance. Basically two

Corresponding authors
Jamil Hussain, jamil@sejong.ac.kr
Sapna Juneja,
sapnajuneja1983@gmail.com

types of feature extraction techniques have been existing. One is automatic feature extraction and other is manual feature extraction. The process of manually extracting features from a text image involves identifying the image's distinctive qualities and putting the extraction techniques into practice. The performance of the models that are working based on manual extraction feature methods is found to be ineffective on degraded documents having text with overlapped characters or text with missed characters. For such situations, deep neural network can be used to automatically extraction of features from text images in place of manual feature extraction and the same has been employed in the present work. Further deep learning can be of two types customized convolution neural networks or pertained transfer learning models. Automatic feature extraction from text images can be accomplished by any from pertained models such as used by *Pramanik, Dansena & Bag (2018)* on ISI Oriya words images or customized convolution neural networks *i.e.*, *Sharma, Gupta & Kumar (2021)* has used on Gurumukhi words images. In the context of deep learning, pertained models are specifically techniques that apply the features learnt by a network on a particular problem to tackle a different set of problems in the same domain. Although pertained models have many advantages, including the ability to save computational time and be particularly helpful for small datasets, they may have significant problems when used to solve different problems in the same domain in which they were trained. These problems include negative transfer and the inability to accurately identify decision boundaries among multiple classes in the target domain's dataset. Therefore, for effective performance in such automatic text recognition system applications, it is advised to construct a custom convolution neural network whose learning would be initialized from scratch. Motivated from the same, the present work has been devised a convolution neural network from scratch to recognize the text written in Gurumukhi. In the next section, a detailed literature review on various text recognition models working using manual feature extraction methods and automatic feature extraction methods has been given to analyze their performance.

## RELATED WORK

*Kumar, Jindal & Sharma (2011)* employed k-nearest neighbor (KNN) and hidden Markov model (HMM) models for Gurumukhi character recognition in order to compare the handwritings of different writers. A radial basis function (RBF)-based support vector machine (SVM) model has been proposed for character recognition by *Kumar, Sharma & Sharma (2014)*. The suggested model has obtained an accuracy of 93.33% on Gurumukhi Mukta. An article by *Kumar, Sharma & Jindal (2014)* employed KNN and SVM classification models with various curve-fitting-based feature extraction techniques such as parabolas and power curves for character recognition in Gurumukhi and gave an accuracy of 98.10%. For the text recognition in Gurumukhi, *Kumar, Jindal & Sharma (2017)* used a support vector machine (SVM) model with different transformations such as discrete wavelet and cosine, fast Fourier, *etc*. Using this approach, testing a dataset of 10,500 character images resulted in 95.8% accuracy. In Gurumukhi script, finite state automata (FSA)-based character recognition was developed by *Singh, Sharma & Singh (2018)*. On 8,200 characters produced by 20 distinct writers, the suggested recognition model resulted

in 97.3% accuracy. *Mahto, Bhatia & Sharma (2018)* explored an SVM classification model with histogram and pyramid gradient features for Gurumukhi character classification. The results demonstrated 99.1% accuracy using the pyramid-oriented gradient features. A novel method of author identification using Gurumukhi's characters is explored by *Kumar et al. (2018)*. This approach, using a SVM model with zone features, transition, and peak extent based (PEB) features on 31.5 million sample Gurumukhi character images, has produced an accuracy of 89.85%. Boosting and bagging classifiers to classify medieval text written in Gurumukhi was explored by *Kumar et al. (2019a)*. When compared to other techniques, the technique achieves recognition accuracy of up to 95.91% by combining classifiers with a voting scheme. Three different classification models named KNN, decision tree (DT) and random forest (RF) have been tested on an unprocessed, unconstrained Gurumukhi character dataset for its 56 different classes of characters by *Garg, Jindal & Singh (2019)*. With the RF classifier in combination with zoning and shadow features, an accuracy of 96.03% was achieved. *Garg, Jindal & Singh (2019)* evaluated the effect of an integrated classifier with features extracted from text on the identification of the Gurumukhi character. Hence, two classifiers named KNN and SVM were employed. The findings demonstrated that linear SVM, in union with polynomial SVM and KNN classifiers, gave 92.3% accuracy.

In their article, *Kumar et al. (2019b)* examined the effectiveness of classifiers for classifying characters and numerals in Gurumukhi script. Using the RF classification technique, accuracy resulted in 87.9% when tested on 13,000 sample images. A CNN-based classification model has been employed for 3,500 Gurumukhi characters by *Jindal et al. (2020)*. The designed model, with two layers for each operation (convolutional and pooling), has achieved 98.32% training accuracy and 74.66% testing accuracy. Similarly, an integrated CNN classification model with a cuckoo search algorithm for optimal feature extraction of Gurumukhi characters has been proposed by *Mushtaq et al. (2021)*. This technique demonstrated that the CNN model can be used to detect characters or text from images of Gurumukhi script with the lowest possible error rate. A deep CNN network has been suggested by *Mahto, Bhatia & Sharma (2021)* for Gurumukhi characters and number recognition. The proposed model developed with one input layer, two repeated convolutional layers, and pooling layers. As per the results analysis, deep CNN is effective at recognizing Gurumukhi characters on the dataset Db1 and numerals on the dataset Db2, as character recognition accuracy for the Db1 dataset is 98.5% and for the Db2 dataset is 98.6%.

Till now, numerous studies have been reported for character recognition in Gurumukhi scripts, but now researchers are putting greater emphasis on word recognition owing to the laborious process of character recognition. However, word recognition is a more arduous task than character recognition but interesting as well. Word recognition can be performed using character level segmentation of words called the analytical technique or using a method devoid of segmentation called the holistic technique. Most of the reported work in Gurumukhi word recognition has been emphasized on using an analytical approach. For example, in their article, *Rani, Dhir & Lehal (2012)*, they proposed an analytical approach to recognize the words written in English and Gurumukhi language. For this, SVM, CNN,

and probabilistic neural network-based classification models were proposed with different features, such as structural features, Gabor features, and Discrete Cosine Transforms (DCT) features. It has been discovered that combining Gabor features with an SVM classifier produces better results than KNN. Similarly, analytical word recognition in Gurumukhi is presented by *Kumar & Gupta (2018)*. Using distinct techniques for feature extraction, including local binary pattern, regional, and directional features, along with an ANN classification model, recognition accuracy resulted in 99.3% on 2,700 Gurumukhi words. *Kaur & Kumar (2021)* employed an analytical approach to Gurumukhi place name recognition. In this work, 40,000 words of place names in Gurumukhi have been recognized using DT and RF classification models. Based on various assessment factors, it was found that the chi-squared attribute (CSA) feature extraction with an RF classifier resulted in 87.42% accuracy.

Word recognition using an analytical approach produces the best results on well-printed or handwritten documents, but it fails to give satisfactory results on degraded documents with cursive handwriting and words with closely written characters, *etc*. For error-free results on those records on which word segmentation is difficult, word recognition using a method devoid of segmentation is employed, called holistic word recognition. The present research work is based on the same for Gurumukhi month's name recognition.

In intending to propose the design of a classification model for an automated month's name recognition system in Gurumukhi, the author has designed a CNN from scratch that recognizes the word without segmentation or using a holistic approach. This model is composed of five layers of convolutional and three layers of max pooling. For an effective performance evaluation of this designed model, a dataset containing 24,000 handwritten word images of Gurumukhi's months for 24 classes was also prepared, to which 500 distinct writers of heterogeneous occupations and age brackets contributed. This model was trained on 24,000 Gurumukhi handwritten word images and became a proficiency model by learning different writing styles in Gurumukhi text. As a result, it has demonstrated exceptional performance.

The subsequent points provide a clear explanation of the primary novelties that result from the current research endeavor in the target domain:

1) In lieu of devising an automated Gurumukhi text recognition system for regional applications, initially a unique dataset of 24,000 words and images belonging to 24 Gurumukhi months's names was prepared from 500 writers of distinct professions and ages.

2) For superlative classification of Gurumukhi months into 24 different classes, a CNN model is devised from scratch.

3) The devised CNN model was designed with the least complex architecture and achieved 99.7% text recognition accuracy on the given dataset, transforming it into a proficiency model by learning different writing styles in Gurumukhi text, which can be used in an automated text classification system for regional applications.

This article is composed of many sections. The introduction and related work are given in "Introduction" and "Related Work". In "Research Methodology", the detailed methodology is demonstrated. The findings and analysis are laid out in "Results Analysis of the Proposed Model". The results are displayed graphically in "Graphical Representation of Results". The performance of the suggested model is contrasted to that of other transfer learning techniques and previously reported text recognition models in "Performance Comparison of Proposed Model". Ultimately, the work's conclusion is provided in the "Conclusion".

## RESEARCH METHODOLOGY

Month's name recognition in Gurumukhi script based on holistic approach has been proposed in this work. Based on this, a handwritten Gurumukhi month's name recognition model has been developed. A dataset of 24,000 handwritten word images of Gurumukhi's months for 24 classes was prepared for this model, to which 500 different authors of various ages and occupations contributed. A research methodology developed to recognize the month's name dataset using a designed CNN model has been shown in Fig. 1.

The research methodology comprises three different stages. The first stage is dataset preparation. In this stage, different steps to dataset preparation have been performed, such as data collection, digitization, gray image conversion, erosion, cropping, and sorting. In the second stage, dataset simulation has been performed. For dataset simulation, a CNN model was devised from scratch. The performance of the model is tested at distinct epochs and batch sizes (BS), at different learning rates, and at different percentages of the dataset. In the final stage, a designed model's performance has been compared with several transfer learning models and existing models for word recognition. The detailed description of all three stages that comprise the research methodology has been given in the following section.

### Dataset preparation

Dataset preparation is the initial phase of the proposed research methodology. A step-by-step procedure for creating a dataset of Gurumukhi months is provided below.

1) A first step to prepare a Gurumukhi month's dataset was to collect a handwritten month's name dataset on a sheet of paper of A4 size. A blank sheet containing 48 blocks on an A4-size sheet has been shown in Fig. 2A.
   This A4-sized sheet contains the handwritten word data of 24 distinct classes of Gurumukhi months written on the same sheet in different blocks. These blocks on the sheet have only one word or one month's name in each block. For each month's class, 1,000 word samples have been collected, for which 500 distinct writers contributed by writing twice the class name on the sheet in two different blocks; thereby, a dataset has been prepared with 24,000 handwritten word samples.

2) The 2nd step was digitization. In this step, digitization of all 500 A4-sized papers from 500 different writers has been completed. A digitized sample sheet from writer 1 has

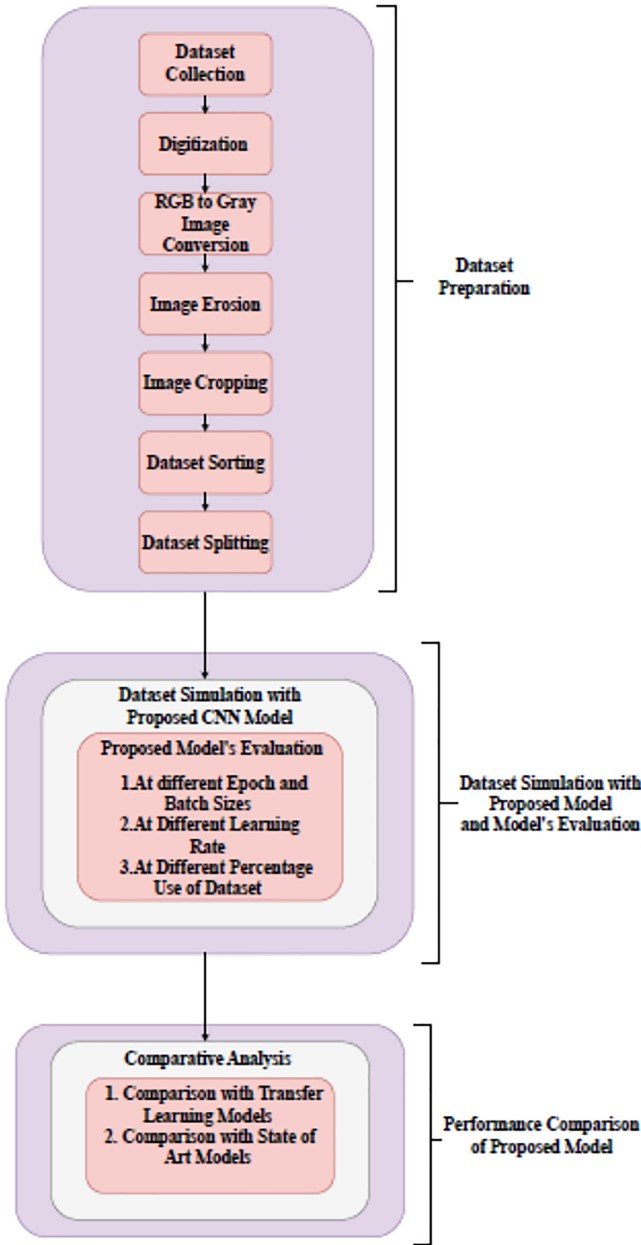

**Figure 1 Block diagram of research methodology.**

been shown in Fig. 2B. A camera of an OPPO (F1s) smart phone has been used for digitization. Every digital copy of a sample sheet has been stored as an image with a resolution of 1,024 × 786 pixels.

3) The 3rd step was RGB to gray image conversion of the digitized dataset. For this, Matlab operation 'rgb2gray' has been used. The resulted image is shown in Fig. 2C.

4) In the 4th step, an eroding operation was performed. This operation was performed on a whole dataset of grayscale images using the MATLAB 'imerode' command. The output of this operation is depicted in Fig. 2D.

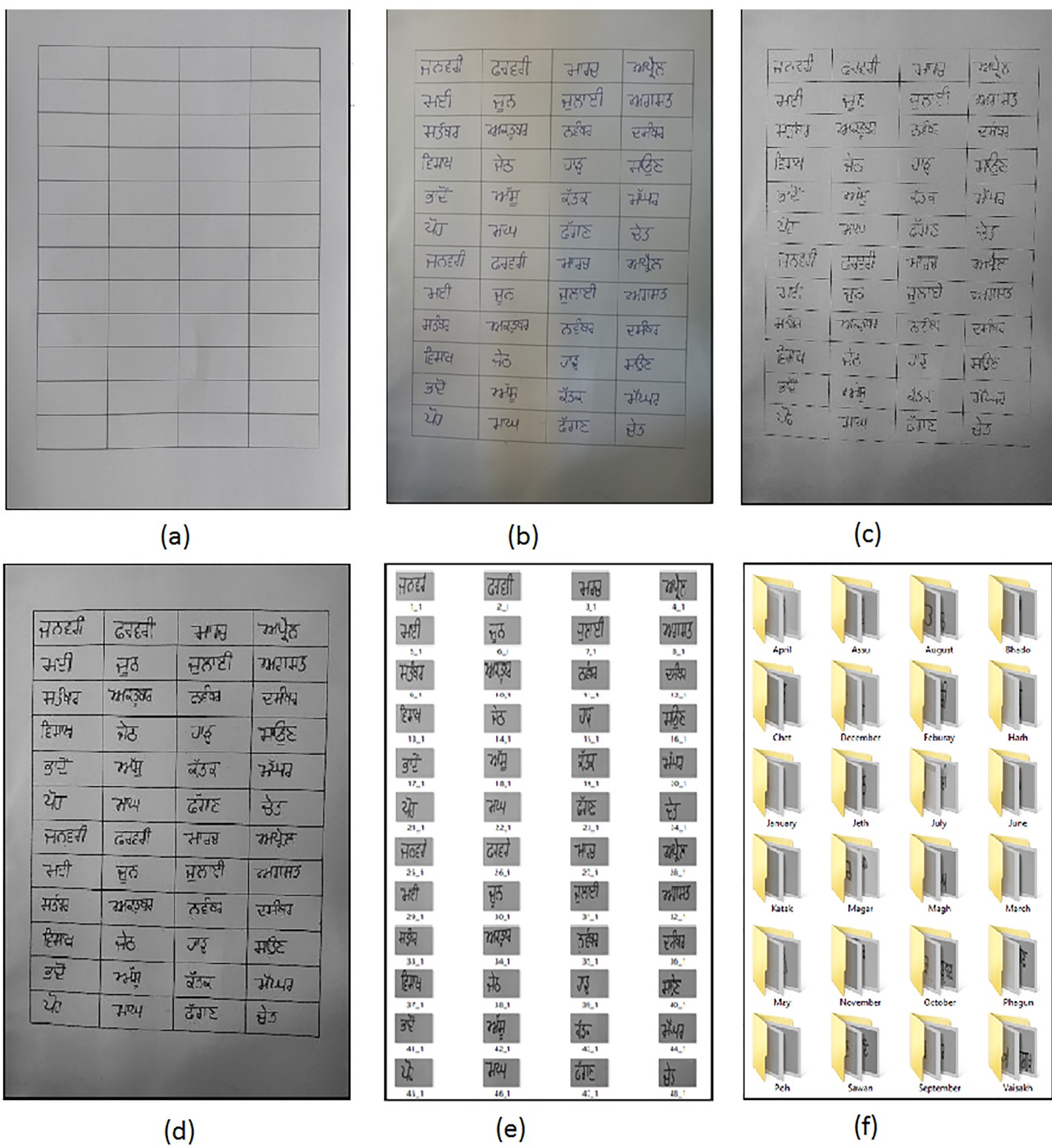

**Figure 2** (A) Void A4 sheet. (B) Handwritten sample of dataset. (C) Color to gray image. (D) Erosion. (E) Cropping. (F) Sorted data in respective folder.

5) In the 5th step, image cropping was done on eroded images using the "imcrop" function of MATLAB, which results in 48 cropped images of each sample erode image as shown in Fig. 2E.

6) Dataset sorting has been done in the 6th step of dataset preparation. This step results in class-name-wise dataset sorting of cropped images. Cropped images for each class name have been saved into the respective folders for each of the 24 classes of the month, as shown in Fig. 2F. This results in a collection of 1,000 images for each class name in the dataset.

7) In the 7th step, dataset splitting has been performed to retain the training and testing sets. The Gurumukhi month's name dataset has been split 80:20. That means, 80% of the dataset, or 19,200 labeled images, from the total of 24,000 images in the Gurumukhi handwritten words dataset was employed for training the model, while the left 20% of the dataset, or 4,800 labeled images, were utilized for the model's testing. The detailed description of the total count of training images and testing images chosen from each Gurumukhi month class is also shown in Table 1.

## Dataset simulation with the proposed model

Present work devised a CNN model entirely from scratch for recognizing the Gurumukhi months. The proposed model will help to recognize the word image of the month's name without segmentation. The self-devised CNN model's description is given in the following section.

### Architecture of the proposed model

The architecture design of the self-devised CNN model can be seen in Fig. 3.

The self-devised CNN model has been composed of five layers of convolution, three layers of max-pooling, and two dense layers. Convolution, activation, pooling and dense layers are the primary layers of the architecture, which help to automatically retrieve the features from a word image of a month's name. Each primary layer in the model has a variable count of filters with its own specific size for automatic feature extraction. Convolutional layers have a fixed size of filter ($3 \times 3$) with a variable count: 32 filters in the first layer, 64 filters map in the 2nd and 3rd layer, and 128 filters map in the 4th and 5th layer. The size of the max pooling layers has varied for the 1st and subsequent layers. The filter size in the 1st layer of maximum pooling is $3 \times 3$, and it is $2 \times 2$ in the 2nd and 3rd layers. The model's outermost layers are dense layers. In order to categorize Gurumukhi word data into 24 classes, the first dense layer employed 1,024 neurons and a ReLu activation function, while the second dense layer uses softmax as an activation function.

Other than these layers, the model has also been composed of two secondary layers known as batch normalization and dropout. For maintaining the output of the network distant from the saturation area, batch normalization has been implemented after each convolutional layer with the variance as well as the mean.

| Class no. | 1 | 2 | 3 | 4 | 5 | 6 |
|---|---|---|---|---|---|---|
| **Table 1** Class wise dataset splitting. | | | | | | |
| Class Name in English | Vaisakh | Jeth | Harh | Sawan | Bhado | Assu |
| Class Name in Gurumukhi | ਵਿਸਾਖ | ਜੇਠ | ਹਾੜ੍ਹ | ਸਾਉਣ | ਭਾਦੋਂ | ਅੱਸੂ |
| Number of Training Samples | 795 | 801 | 799 | 782 | 802 | 809 |
| Number of Testing Samples | 205 | 199 | 201 | 218 | 198 | 191 |
| Class no. | 7 | 8 | 9 | 10 | 11 | 12 |
| Class Name in English | Katak | Magar | Poh | Magh | Phagun | Chet |
| Class Name in Gurumukhi | ਕੱਤਕ | ਮੱਘਰ | ਪੋਹ | ਮਾਘ | ਫੱਗਣ | ਚੇਤ |
| Number of training samples | 813 | 789 | 798 | 788 | 802 | 819 |
| Number of testing samples | 187 | 211 | 202 | 212 | 198 | 181 |
| Class no. | 13 | 14 | 15 | 16 | 17 | 18 |
| Class name in English | January | February | March | April | May | June |
| Class name in Gurumukhi | ਜਨਵਰੀ | ਫਰਵਰੀ | ਮਾਰਚ | ਅਪ੍ਰੈਲ | ਮਈ | ਜੂਨ |
| Number of training samples | 770 | 817 | 804 | 786 | 803 | 822 |
| Number of testing samples | 230 | 183 | 196 | 214 | 197 | 178 |
| Class no. | 19 | 20 | 21 | 22 | 23 | 24 |
| Class name in English | July | August | September | October | November | December |
| Class name in Gurumukhi | ਜੁਲਾਈ | ਅਗਾਸਤ | ਸਤੰਬਰ | ਅਕਤੂਬਰ | ਨਵੰਬਰ | ਦਸੰਬਰ |
| Number of training samples | 797 | 788 | 805 | 804 | 808 | 799 |
| Number of testing samples | 203 | 212 | 195 | 196 | 192 | 201 |

To avoid overfitting, the model has deployed a dropout layer with a 0.25 value. As a result of this, 25% of the neurons are momentarily and randomly eliminated. When combined, these layers create an automatic feature extractor for the model. Table 2 provides a detailed explanation of the layers and the learning parameters of the self-devised CNN model.

### Experiment layout

The design of the self-devised CNN model was built using Keras, with TensorFlow backend libraries available in Python. Further, it has been tested on Google Colab for Gurumukhi month's name recognition using Google Colab's Tesla K80 GPU, which has a maximum limit of 12 h of continuous use and RAM and ROM of 12.69 gigabytes and 107.72 gigabytes, respectively.

The value of various learning parameters used for the proposed CNN model has been given below in Table 3.

## RESULTS ANALYSIS OF THE PROPOSED MODEL

The proposed model was used for recognizing the Gurumukhi months. The performance of the designed model is tested at different epochs and BS, at different learning rates, and at different percentages of the dataset. The detailed analysis of results obtained for the proposed model has been shown in the following sub section.
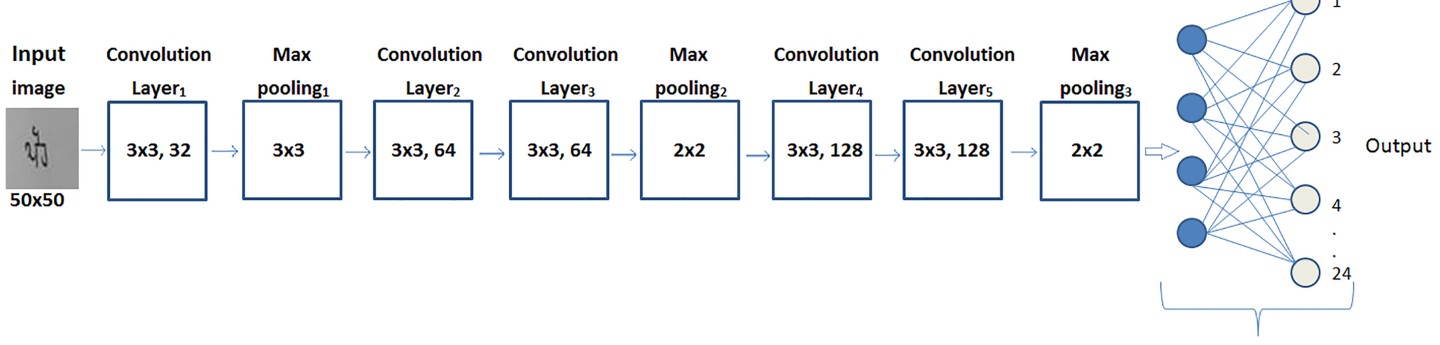

**Figure 3 Architectural framework of the proposed model.**

**Table 2 Description of self-devised CNN model.**

| Type of layers | Output shape | Parameters |
|---|---|---|
| "sequential" | | |
| conv2d (Conv2D) | (None, 50, 50, 32) | 320 |
| activation (Activation) | (None, 50, 50, 32) | 0 |
| batch normalization (Batch Normalization) | (None, 50, 50, 32) | 128 |
| max_pooling2d (MaxPooling2D) | (None, 16, 16, 32) | 0 |
| dropout (Dropout) | (None, 16, 16, 32) | 0 |
| conv2d_1 (Conv2D) | (None, 16, 16, 64) | 18,496 |
| activation_1 (Activation) | (None, 16, 16, 64) | 0 |
| batch_normalization_1 (Batch Normalization) | (None, 16, 16, 64) | 256 |
| conv2d_2 (Conv2D) | (None, 16, 16, 64) | 36,928 |
| activation_2 (Activation) | (None, 16, 16, 64) | 0 |
| batch_normalization_2 (Batch Normalization) | (None, 16, 16, 64) | 256 |
| max_pooling2d_1 (Max-Pooling 2D) | (None, 8, 8, 64) | 0 |
| dropout_1 (Dropout) | (None, 8, 8, 64) | 0 |
| conv2d_3 (Conv2D) | (None, 8, 8, 128) | 73,856 |
| activation_3 (Activation) | (None, 8, 8, 128) | 0 |
| batch normalization_3 (Batch Normalization) | (None, 8, 8, 128) | 512 |
| conv2d_4 (Conv2D) | (None, 8, 8, 128) | 147,584 |
| activation_4 (Activation) | (None, 8, 8, 128) | 0 |
| batch normalization_4 (Batch Normalization) | (None, 8, 8, 128) | 512 |
| max_pooling2d_2 (MaxPooling2D) | (None, 4, 4, 128) | 0 |
| dropout_2 (Dropout) | (None, 4, 4, 128) | 0 |
| flatten (Flatten) | (None, 2,048) | 0 |
| dense (Dense) | (None, 1,024) | 2,098,176 |
| activation_5 (Activation) | (None, 1,024) | 0 |
| batch_normalization_5 (Batch Normalization) | (None, 1,024) | 4,096 |
| dropout_3 (Dropout) | (None, 1,024) | 0 |

| Type of layers | Output shape | Parameters |
|---|---|---|
| dense_1 (Dense) | (None, 24) | 24,600 |
| activation_6 (Activation) | (None, 24) | 0 |
| Total params: 2,405,720 | | |
| Trainable params: 2,402,840 | | |
| Non-trainable params: 2,880 | | |

**Table 3 Learning parameters details.**

| Network type | Optimizer | Hidden layer activation function | Loss function | Matrix | No. of epochs | No. of BS |
|---|---|---|---|---|---|---|
| Feed forward back propagation | Adam, LR = 0.001, $\beta_1$ = 0.9, $\beta_2$ = 0.999, epsilon = 0.0000001, decay = LR/Epoch | ReLu and Softmax | Categorical cross entropy | Validation accuracy | 100 | 40 |

## Analysis at different epochs and BS

Initially, the results of the self-devised CNN model have been obtained at different epochs and different BS for various learning parameters and confusion matrix parameters. Table 4 presents the performance of the CNN model at 100, 80, 60, and 40 epochs with 20, 30, and 40 BS.

Table 4 analysis shows that, when the model has been simulated on 100 epochs with 40 BS, the maximum training and testing accuracy has been obtained, whose values are 99% and 99.73%, respectively. On the other hand, the lowest testing accuracy (89.85%) has been observed for 20 BS and 40 epochs.

On the other hand, the results analysis of precision, recall, and F1 score has shown that the self-devised model has given its best score for each of these parameters also on 100 epochs and 40 BS. The highest values of precision, recall, and F1 score on 100 epochs with 40 BS are 0.9974, 0.9973, and 0.9973, respectively.

## Analysis at different learning rates

Here the results of the self-devised CNN model on the Gurumukhi month's name dataset have been analyzed at different learning rates. The learning rates used for testing the proposed model range from 1e−1 to 1e−6. Table 5 gives detailed results analyses of the proposed model for various performance matrices and learning parameter values.

Table 5 shows that at the highest learning rate (LR = 1e−1), the model's performance was poor for each performance matrix, but as the LR was reduced to 1e−3 from 1e−1, the model's performance improved to the best value for each performance matrix, and a further decrease in the LR degraded the model's results.

The value of maximum training and testing accuracy along with precision, recall, and F1 score on LR = 1e−3 is 99% and 99.73%, 0.9974, 0.9973, and 0.9973, respectively, whereas on LR = 1e−1, training accuracy, testing accuracy, F1 score, recall, and precision are 0.0996, 0.1300, 0.7301, 0.1290, and 0.2086, respectively.

**Table 4 Proposed model's performance at different epochs and batch sizes.**

| Parameters | Epoch = 40 | | | Epoch = 60 | | | Epoch = 80 | | | Epoch = 100 | | |
|---|---|---|---|---|---|---|---|---|---|---|---|---|
| Results | BS = 20 | BS = 30 | BS = 40 | BS = 20 | BS = 30 | BS = 40 | BS = 20 | BS = 30 | BS = 40 | BS = 20 | BS = 30 | BS = 40 |
| Training accuracy | 0.9486 | 0.9550 | 0.9568 | 0.9604 | 0.9649 | 0.9679 | 0.9696 | 0.9727 | 0.9738 | 0.9703 | 0.9763 | 0.99 |
| Validation accuracy | 0.8985 | 0.9765 | 0.9885 | 0.9937 | 0.9933 | 0.9917 | 0.9958 | 0.9948 | 0.9877 | 0.9950 | 0.9908 | 0.9973 |
| Training loss | 0.1580 | 0.1382 | 0.1440 | 0.1231 | 0.1061 | 0.0939 | 0.0974 | 0.0883 | 0.0792 | 0.0885 | 0.0749 | 0.0313 |
| Validation loss | 1.0888 | 0.0754 | 0.0404 | 0.0251 | 0.0271 | 0.0363 | 0.0249 | 0.0277 | 0.0475 | 0.0230 | 0.0359 | 0.0188 |
| Precision | 0.9116 | 0.9802 | 0.9887 | 0.9937 | 0.9935 | 0.9918 | 0.9958 | 0.9948 | 0.9880 | 0.9950 | 0.9911 | 0.9974 |
| Recall | 0.8974 | 0.9777 | 0.9887 | 0.9937 | 0.9933 | 0.9914 | 0.9958 | 0.9948 | 0.9877 | 0.9950 | 0.9912 | 0.9973 |
| F1 score | 0.8994 | 0.9770 | 0.9886 | 0.9937 | 0.9934 | 0.9917 | 0.9958 | 0.9948 | 0.9876 | 0.9951 | 0.9909 | 0.9973 |

**Table 5 Analysis of proposed model at different learning rate.**

| Parameters | Learning rate | | | | | |
|---|---|---|---|---|---|---|
| | Epoch = 60 | | | | | |
| Results | LR = 1e−1 | LR = 1e−2 | LR = 1e−3 | LR = 1e−4 | LR = 1e−5 | LR = 1e−6 |
| Training accuracy | 0.0996 | 0.9548 | 0.99 | 0.9815 | 0.7970 | 0.1737 |
| Validation accuracy | 0.1300 | 0.9933 | 0.9973 | 0.9940 | 0.8815 | 0.2994 |
| Training loss | 29.74 | 0.2646 | 0.0313 | 0.0556 | 0.6253 | 3.1811 |
| Validation loss | 2,545 | 0.0874 | 0.0188 | 0.0379 | 0.3387 | 2.3097 |
| Precision | 0.2086 | 0.9932 | 0.9974 | 0.9940 | 0.9013 | 0.9379 |
| Recall | 0.1290 | 0.9934 | 0.9973 | 0.9938 | 0.8791 | 0.3012 |
| F1 score | 0.7301 | 0.9933 | 0.9973 | 0.9938 | 0.8814 | 0.2903 |

## Analysis at different percentage use of dataset

Finally, the suitability of the self-devised CNN model has been evaluated using distinct percentages of the dataset. In this analysis, the proposed model was tested on 30% of the dataset, 50% of the dataset, and 100% of the entire dataset. The total images in the dataset are 24,000; thus, 30% and 50% of the dataset contained 7,200 and 12,000 images, respectively. While performing the analysis using 30% of the dataset, the dataset for training and testing was split into 80:20. That means from 30% (7,200 images) of the whole dataset, 5,760 images were used for training the model, and 1,840 images were used for testing the model. Similarly, while performing the analysis using 50% (12,000 images) of the dataset, the dataset for training and testing was again split into 80:20. That means 9,600 images were used for training the model, and 2,400 images were used for testing the model. Further, while using the whole dataset (100% or 24,000 images), according to the 80:20 split, 19,200 images were used for the model's training and 4,800 images were used for testing. Table 6 provides detailed results analysis with various percentages of use of the dataset.

**Table 6 Analysis of proposed model at different percentage use of dataset.**

| Results | Dataset percentage Epoch = 60 | | |
|---|---|---|---|
| | DS = 30% | DS = 50% | DS = 100% |
| Training accuracy | 0.9696 | 0.9806 | 0.99 |
| Validation accuracy | 0.8354 | 0.9829 | 0.9973 |
| Training loss | 0.0935 | 0.0637 | 0.0313 |
| Validation loss | 0.7428 | 0.0692 | 0.0188 |
| Precision | 0.8891 | 0.9841 | 0.9974 |
| Recall | 0.8342 | 0.9834 | 0.9973 |
| F1 score | 0.8315 | 0.9832 | 0.9973 |

Table 6 demonstrates when the model has been simulated using 30%, 50%, and 100% of the dataset. It can be seen that the model's results at 100% of the dataset outperformed the results obtained at 30% and 50% of the dataset.

The model on 30% of the dataset results in a maximum testing accuracy of 83.54%, the model on 50% of the dataset results in a testing accuracy of 98.29%, and the model on the entire dataset (100%) results in a testing accuracy of 99.73%.

Hence, the CNN model on an entire dataset at 100 epoch with 40 BS and LR = 1e−3 performed exceptionally well in terms of each performance evaluation matrix, including accuracy, precision, F1 score, and recall. The graphical representation of results in terms of accuracy and loss curves, along with a confusion matrix for the model at 100 epochs with 40 BS using LR = 1e−3, have been presented in the next section.

## GRAPHICAL REPRESENTATION OF RESULTS

It was found from the previous section that, the self-devised CNN model on an entire dataset at 100 epoch with 40 BS using LR = 1e−3 performed best in term of training parameters results and confusion matrix results. The graphical representation of the same has been presented in this section.

### Accuracy graph

The graph for accuracy and loss for the self-devised CNN model at 100 epochs and 40 BS with LR = 1e−3 is shown in Fig. 4. Figure 4 depicts that with the increase in accuracy, the loss of the proposed model has decreased. As per the analysis of accuracy and loss curves for the model, the maximum training and testing loss at the first epoch is 3.3248 and 2.0870, respectively, which has been decreased to 0.0313 for training loss and 0.0188 for testing loss when reaching 100 epochs.

Similarly, at 100 epochs, the final training and testing accuracy have been reached at 99% and 99.73%, respectively, from 16.32% for training accuracy and 35.29% for testing accuracy at the first epoch.
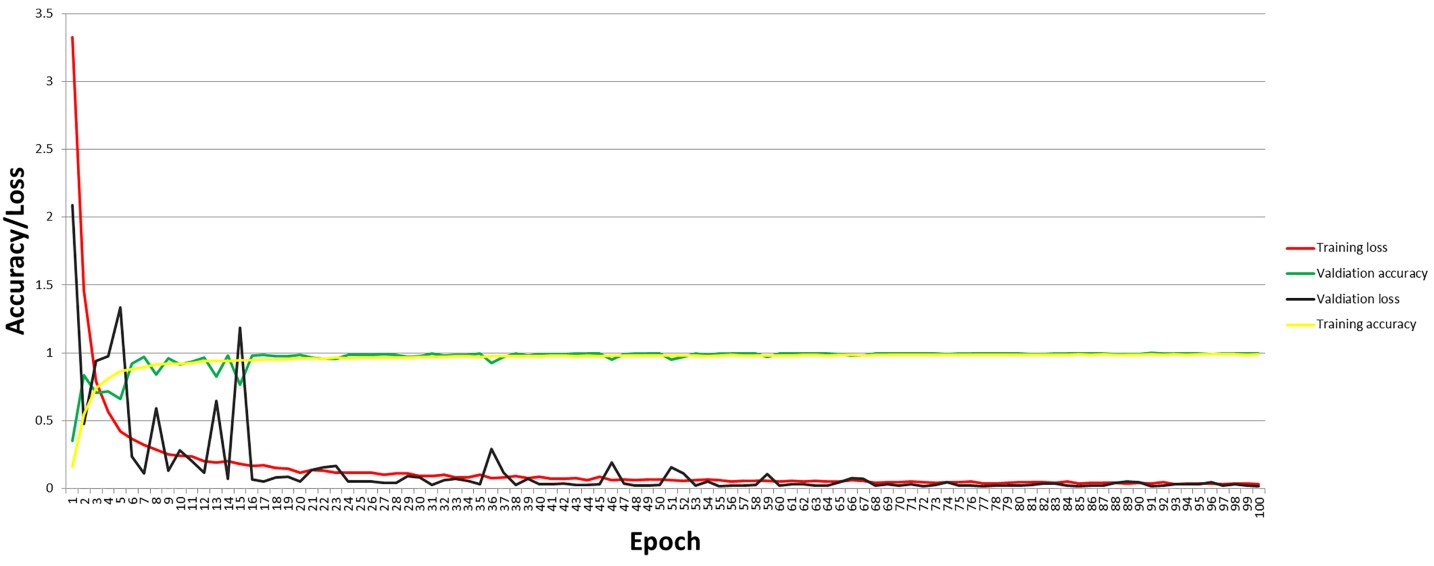

**Figure 4** **Accuracy and loss curve of the proposed model.**

## Confusion matrix

The confusion matrixes of the self-devised CNN model on 100 and 40 BS with LR = 1e−3 have been presented in Fig. 5. The matrix gives a clear and succinct summary of both correct and incorrect predictions. The names of the classes have been appearing in every column as well as the row that belongs to the matrix. The diagonal values of matrices show the precise count of text images of Gurumukhi month's name that a given model has correctly classified.

From Fig. 5, the results for each performance assessment parameter, including precision, recall, and F1 score, have been presented for this overall score and class-wise score in Table 7 at 100 epochs and 40 BS.

It has been analyzed from the Table 7 that, the value of precision is resulting in 1 for 'ਅਪ੍ਰੈਲ', 'ਅੱਸੂ', 'ਅਗਸਤ', 'ਚੇਤ', 'ਜਨਵਰੀ', 'ਜੂਨ', 'ਕੱਤਕ', 'ਨਵੰਬਰ', 'ਫੱਗਣ', 'ਪੋਹ', and 'ਸਤੰਬਰ', 'ਅਪ੍ਰੈਲ', 'ਅੱਸੂ', 'ਅਗਸਤ', 'ਚੇਤ', 'ਦਸੰਬਰ', 'ਫਰਵਰੀ', 'ਜੁਲਾਈ', 'ਜੇਠ', 'ਜੂਨ', 'ਕੱਤਕ', 'ਮੱਘਰ', 'ਮਾਰਚ', 'ਨਵੰਬਰ', 'ਅਕਤੂਬਰ', 'ਫੱਗਣ' and 'ਪੋਹ' month's name. F1 score has maximum value of 1 which is for 'ਅਪ੍ਰੈਲ', 'ਅੱਸੂ', 'ਜੂਨ', 'ਮੱਘਰ', 'ਮਾਰਚ', 'ਨਵੰਬਰ', 'ਅਕਤੂਬਰ', 'ਫੱਗਣ' and 'ਪੋਹ' classes. The minimum value of precision and F1 score is around 0.9899 for 'ਮਈ' month. Recall score one for 'ਅਪ੍ਰੈਲ', 'ਅੱਸੂ', 'ਭਾਦੋਂ', 'ਹਾੜ੍ਹ', 'ਜਨਵਰੀ', 'ਜੂਨ', 'ਕੱਤਕ', 'ਮੱਘਰ', 'ਮਾਘ', 'ਮਾਰਚ', 'ਮਈ', 'ਨਵੰਬਰ', 'ਫੱਗਣ' and 'ਪੋਹ' classes.

## Performance comparison of proposed model

This section compares the self-devised CNN model's performance to a various transfer learning models as well as the existing models. The section that follows provides a thorough comparison analysis.

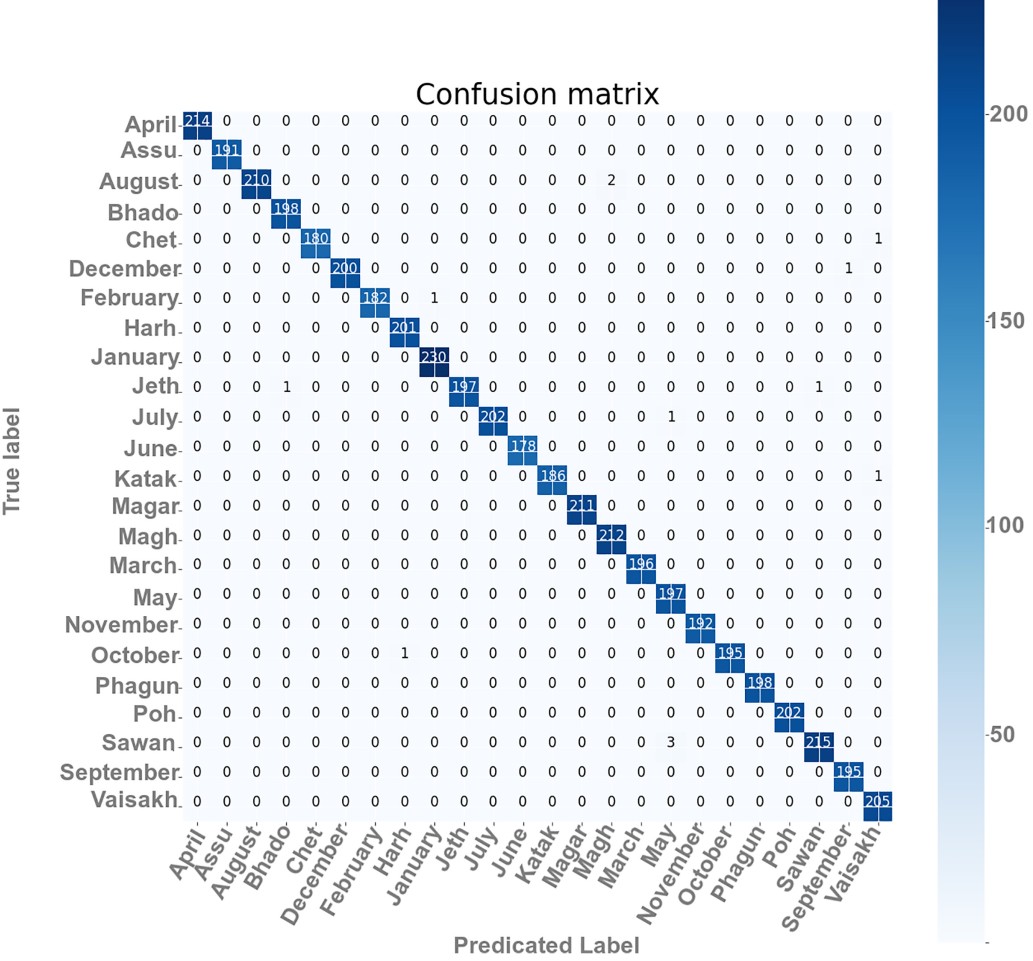

**Figure 5 Confusion matrix of the proposed model.**

**Table 7 Confusion matrix results of proposed CNN model.**

| Class name | Precision | Recall | F1 score |
|---|---|---|---|
| ਅਪ੍ਰੈਲ | 1.0000 | 1.0000 | 1.0000 |
| ਅੱਸੂ | 1.0000 | 1.0000 | 1.0000 |
| ਅਗਸਤ | 1.0000 | 0.9906 | 0.9953 |
| ਭਾਦੋਂ | 0.9950 | 1.0000 | 0.9975 |
| ਚੇਤ | 1.0000 | 0.9945 | 0.9972 |
| ਦਸੰਬਰ | 1.0000 | 0.9950 | 0.9975 |
| ਫਰਵਰੀ | 1.0000 | 0.9945 | 0.9973 |
| ਹਾੜ੍ਹ | 0.9950 | 1.0000 | 0.9975 |
| ਜਨਵਰੀ | 0.9957 | 1.0000 | 0.9978 |
| ਜੇਠ | 1.0000 | 0.9899 | 0.9949 |
| ਜੁਲਾਈ | 1.0000 | 0.9951 | 0.9975 |
| ਜੂਨ | 1.0000 | 1.0000 | 1.0000 |
| ਕੱਤਕ | 1.0000 | 0.9947 | 0.9973 |

*(Continued)*

| Class name | Precision | Recall | F1 score |
|---|---|---|---|
| ਮੱਘਰ | 1.0000 | 1.0000 | 1.0000 |
| ਮਾਘ | 0.9907 | 1.0000 | 0.9953 |
| ਮਾਰਚ | 1.0000 | 1.0000 | 1.0000 |
| ਮਈ | 0.9801 | 1.0000 | 0.9899 |
| ਨਵੰਬਰ | 1.0000 | 1.0000 | 1.0000 |
| ਅਕਤੂਬਰ | 1.0000 | 0.9949 | 0.9974 |
| ਫੱਗਣ | 1.0000 | 1.0000 | 1.0000 |
| ਪੋਹ | 1.0000 | 1.0000 | 1.0000 |
| ਸਾਉਣ | 0.9954 | 0.9862 | 0.9908 |
| ਸਤੰਬਰ | 0.9949 | 1.0000 | 0.9974 |
| ਵਿਸਾਖ | 0.9903 | 1.0000 | 0.9951 |

**Table 8 Comparison between different models.**

| Parameters<br>Model | Training accuracy | Validation accuracy | Training loss | Validation loss | Precision | Recall | F1 score |
|---|---|---|---|---|---|---|---|
| ResNet-50 | 0.3335 | 0.4015 | 2.1658 | 1.8842 | 0.4528 | 0.4021 | 0.3967 |
| MobileNet | 0.7108 | 0.7108 | 0.8631 | 0.8631 | 0.7384 | 0.7094 | 0.7117 |
| VGG-19 | 0.7452 | 0.7758 | 0.7693 | 0.6744 | 0.7908 | 0.7756 | 0.7753 |
| VGG-16 | 0.7832 | 0.8056 | 0.6515 | 0.5720 | 0.8173 | 0.8042 | 0.8023 |
| EfficientNet | 0.7995 | 0.7996 | 0.6033 | 0.6035 | 0.8207 | 0.79822 | 0.7999 |
| DenseNet-121 | 0.8124 | 0.8256 | 0.5614 | 0.5181 | 0.8340 | 0.8264 | 0.8271 |
| **Proposed Model** | **0.9703** | **0.9950** | **0.0885** | **0.0230** | **0.9950** | **0.9951** | **0.9950** |

**Note:**
Bold values indicate instances where the proposed model achieves higher performance compared to the baseline models.

## Comparison with transfer learning models

This section compared the self-devised CNN model's performance with distinct transfer learning models for word recognition. For this analysis, four transfer learning models have been chosen. A comparison between the models in terms of various parameters has been shown in Table 8.

Table 8 shows that the proposed model for 24 classes' classification of the Gurumukhi handwritten month's names dataset at 100 epochs and 40 BS outperformed transfer learning models for each evaluation parameter. In contrast to this, ResNet-50 falls behind all other models for the same classification problem.

## Comparison with existing models

Here the effectiveness of the self-devised CNN model for text recognition in Gurumukhi has been compared with that of the existing models available. Table 9 demonstrates the model's comparison in terms of technique, dataset, and results.

**Table 9 Proposed model comparison with existing models.**

| References/Year | Technique used | Dataset used | Results |
|---|---|---|---|
| *Rani, Dhir & Lehal (2012)* | Gabor features with an SVM classifier | 11,400 words images | 99.402% |
| *Sharma et al. (2017)* | VGG M architecture with faster-RCNN | 2,300 images of multilingual Indian postal documents | Average precision = 0.569 |
| *Kumar & Gupta (2018)* | ANN with local binary pattern directional and regional features | 2,700 word images | 99.3% |
| *Bhowmik et al. (2019)* | SVM classifier | 18,000 images of handwritten Bangla cities name | Accuracy = 83.64% |
| *Pramanik, Dansena & Bag (2018)* | Alexnet model | ISI Oriya, CMATERdb 3.1.1, 3.1.2 and 3.4.1 | Accuracy = 94.26% |
| *Das et al. (2020)* | H-WordNet model | 18,000 images of handwritten Bangla cities name | Accuracy = 96.17% |
| *Sharma, Gupta & Kumar (2021)* | Proposed CNN model | 22,000 images of district names of Punjab state | Accuracy = 99% |
| *Kaur & Kumar (2021)* | Random forest classifier with Chi-Squared Attribute (CSA) features | 40,000 words images | 87.42% |
| *Rajesh et al. (2022)* | CNN-BiLSTM Network | IAM Handwriting | Accuracy = 89.05% |
| *Salunke et al. (2021)* | Proposed CNN model | 2,000 images of Devanagari word | Accuracy = 94% |
| *Lomte & Doye (2022)* | Proposed CNN model | 70,000 images of Vedic Sanskrit words | Accuracy = 97.42% |
| **Proposed model** | **CNN** | **24,000 handwritten word images of Gurumukhi months** | **99.73%** |

Note:
The bold entries highlight the instances where the proposed method achieves higher performance compared to the related work.

As presented in Table 9, the self-devised CNN model for text classification in regional languages produced an accuracy of 99.25% that is highest reported accuracy among reported models available in a similar context. Also, the dataset utilized for the current task of classification is brand new, exclusive, and not available online or offline.

## CONCLUSION

For automated text recognition in Gurumukhi, a Gurumukhi month's name dataset and CNN-based model have been developed. The model has been devised for five convolutional, three pooling, and two dense layers. On the other hand, a dataset of the month's name in Gurumukhi language has been prepared for 24,000 images. The model's performance on the given dataset of the month's name was tested at distinct epochs and BS, at distinct learning rates, and using different percentages of the dataset. It has been observed that on an entire dataset of 24,000 images, at 100 epoch and 40 BS, with LR = 1e−3, the model gives the best performance. At last, the model's performance was compared with various transfer learning models and other existing models for text classification. Currently, the proposed CNN model was only evaluated for its performance on a single dataset of Gurumukhi text. In the future, another publicly available dataset of Gurumukhi text will be employed to test the model's performance.

### Funding

This work was supported by Institute of Information & Communications Technology Planning & Evaluation (IITP) grant funded by the Korea Government (MSIT) IITP-2017-0-00655, Lean UX Core Technology and platform for any digital artifacts UX evaluation. The funders had no role in study design, data collection and analysis, decision to publish, or preparation of the manuscript.

### Grant Disclosures

The following grant information was disclosed by the authors:
Korea Government (MSIT): IITP-2017-0-00655.
Lean UX Core Technology.

### Competing Interests

The authors declare that they have no competing interests.

### Author Contributions

- Tajinder Pal Singh conceived and designed the experiments, performed the experiments, performed the computation work, authored or reviewed drafts of the article, and approved the final draft.
- Sheifali Gupta conceived and designed the experiments, analyzed the data, performed the computation work, authored or reviewed drafts of the article, and approved the final draft.
- Jamil Hussain analyzed the data, prepared figures and/or tables, authored or reviewed drafts of the article, and approved the final draft.
- Sapna Juneja analyzed the data, prepared figures and/or tables, authored or reviewed drafts of the article, and approved the final draft.
- Meenu Garg performed the experiments, analyzed the data, performed the computation work, prepared figures and/or tables, and approved the final draft.
- Deepali Gupta conceived and designed the experiments, prepared figures and/or tables, authored or reviewed drafts of the article, and approved the final draft.
- Majed Alsafyani performed the computation work, prepared figures and/or tables, authored or reviewed drafts of the article, and approved the final draft.

### Data Availability

   The code and data are available in the Supplemental Files.

### Supplemental Information

Supplemental information for this article can be found online at http://dx.doi.org/10.7717/peerj-cs.2627#supplemental-information.

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
