# Peer review of "Convolutional neural network (CNN) based month’s name recognition in Gurumukhi script for Punjab state of India"

_PeerJ Computer Science, doi:10.7717/peerj-cs.2627_

## Round 0.1 · original submission · Major Revisions

I consider the research is well-defined. The article is an original one and aligns well with the journal's aims and scope.

The authors should mention and justify the novelty of the methodology proposed.

The paper must be improve according to all observations introduced by reviewers.

Reviewer 1 ·

Basic reporting

⦁ In the abstract the authors should mention the test accuracy not the validation accuracy.
⦁ There are some typographical errors in the paper.
⦁ Line (186) remove full stop.
⦁ In the paragraph starting from Line (289), most of the words are not visible, before submitting the manuscript make sure to proofread it.
⦁ In Table 1 also the "Class Name in Gurumukhi" are not visible.
⦁ In Table 7 also the "Class Name" are not visible.

⦁ The language needs to be improved significantly. It is noted that your manuscript needs careful editing by someone with expertise in technical English editing paying particular attention to English grammar, and sentence structure so that the goals and results of the study are clear to the reader.

⦁ Abbreviations and acronyms are often defined the first time they are used within the main text and then used throughout the remainder of the manuscript. Please consider adhering to this convention. Moreover, remove personal pronounces from text.
⦁ Improve the quality of Fig. 4, the text is looking blur.
⦁ There is also a lack of comparison of the proposed method with the recent state-of-art works. So References also need to be updated with more recent works.
⦁ What are some of the limitations of this work? How do you envision it being extended in the future?

Experimental design

No comment

Validity of the findings

⦁ In Section 4.3, the author has evaluated the performance of the model on different percentages of the dataset. But it is totally ambiguous how the author is splitting the data and using the dataset in these ratios. For example when the author is using 100% data for testing then how the training has been performed (as test data should never be touched before the final evaluation). Justify this.

⦁ There is difference between the Validation Accuracy and the Test Accuracy but the authors have used these terms interchangeably. Technically, a test set should (putting it loosely) never be touched before the final evaluation (and a validation set is usually created by splitting the training set). I hope this has been enforced in the work. Please justify this.

Annotated reviews are not available for download in order to protect the identity of reviewers who chose to remain anonymous.

Reviewer 2 ·

Basic reporting

• The English used is poor and should be improved. Some of the sentences can be rewritten, as they are not in a correct form or as they are repeated in the document. Some of such statements have been highlighted.
• There are many typos so the authors have to correct those typos.
• The Introduction and Related Work should be two separate sections in the manuscript.
• The number of references used for the work should be increased.

Experimental design

• Why such a methodology is adopted is not clear. For example,
a) Why the authors have taken batch sizes of 20, 30 and 40 and not 16, 32, 64, etc. which are more standard and adopted widely?
b) While tuning the hyperparameters, which hyperparameters are tuned in which order should also be mentioned and justified.

Validity of the findings

• The authors can add two more SOTA transfer learning models namely EfficientNet and MobileNet which are more efficient and have shown very good results on text recognition, and analyse their performance against the developed dataset.
• How are the accuracies obtained should be clearly mentioned. For example, whether it was by training and testing the models once or by doing so for a certain number of times and taking an average of the accuracies achieved, etc.
• Table 9 shows a comparison of the proposed work with other existing works. However, the works reported in the table had been carried out on different datasets. Therefore, it is not fair to make a comparison between those studies. The authors should check if there are works reported on, if not the same

Additional comments

The work reported is a CNN-based word recognition of Gurumukhi months names. Many works have been reported for word recognition and as such there is not much novelty in the proposed work. The authors should mention and justify the novelty of the methodology proposed.

The manuscript may be considered for publication if the authors address all the comments clearly and do the required major revision.

Annotated reviews are not available for download in order to protect the identity of reviewers who chose to remain anonymous.

---

## Round 0.2 · Major Revisions

The authors must address the suggestions indicated by Reviewer 2.

Reviewer 2 ·

Basic reporting

The introduction is too short. It should be further elaborated to provide readers with a clearer context and a more comprehensive understanding of the problem being addressed.

Experimental design

No comment

Validity of the findings

Addition of EfficientNet and MobileNet in the analysis part is expected from the authors.

---

## Round 0.3 · Major Revisions

The authors must address the comments to the reviewer very clear!

Reviewer 2 ·

Basic reporting

NIL

Experimental design

NIL

Validity of the findings

I am still to find the performance analysis of EfficientNet and MobileNet as it is not available in Table 8 as the authors claimed.

Additional comments

When referring to/acknowledging/ pointing out certain research carried out in the past, the authors should provide the corresponding references. In the introduction section, they can add references of the recent works carried out on regional scripts using CNN and transfer learning.

---

## Round 0.4 · accepted · Accept

The article is well improved!